# Safety and Efficacy of Stand-Alone Bioactive Glass Injectable Putty or Granules in Posterior Vertebral Fusion for Adolescent Idiopathic and Non-Idiopathic Scoliosis

**DOI:** 10.3390/children10020398

**Published:** 2023-02-17

**Authors:** Aurélien Courvoisier, Marie-Christine Maximin, Alice Baroncini

**Affiliations:** 1TIMC, University Grenoble Alpes, 38000 Grenoble, France; 2Grenoble Alps Scoliosis and Spine Center, Grenoble Alps University Hospital, 38043 Grenoble, France; 3Department of Orthopaedics, RWTH Uniklinik Aachen, 52074 Aachen, Germany

**Keywords:** scoliosis, biomaterials, spine, fusion, bioactive glass

## Abstract

Posterior spinal fusion (PSF) is the standard procedure for the treatment of severe scoliosis. PSF is a standard procedure that combines posterior instrumentation with bone grafting and/or bone substitutes to enhance fusion. The aim of this retrospective study was to evaluate and compare the post-operative safety and efficiency of stand-alone bioactive glass putty and granules in posterior spine fusion for scoliosis in a paediatric cohort. A total of 43 children and adolescents were included retrospectively. Each patient’s last follow-up was performed at 24 months and included clinical and radiological evaluations. Pseudarthrosis was defined as a loss of correction measuring >10° of Cobb angle between the pre-operative and last follow-up measurements. There was no significant loss of correction between the immediate post-operative timepoint and the 24-month follow-up. There was no sign of non-union, implant displacement or rod breakage. Bioactive glass in the form of putty or granules is an easily handled biomaterial but still a newcomer on the market. This study shows that the massive use of bioactive glass in posterior fusion, when combined with proper surgical planning, hardware placement and correction, is effective in providing good clinical and radiological outcomes.

## 1. Introduction

Scoliosis is defined as three-dimensional structural deformity of the spine in the anterior-posterior, sagittal and transverse planes. The most common type is adolescent idiopathic scoliosis (AIS), but neurologic or muscular disorders may also lead to progressive spine deformities (non-idiopathic scoliosis, or NS) [1].

In the most severe cases, progression of the deformity necessitates surgery to correct the spinal curvature, rebalance the spine and, above all, stop progression [2,3]. Posterior spinal fusion (PSF) is the standard procedure for the treatment of scoliosis [4]. In paediatrics, this surgery improves self-esteem and general appearance [3]. PSF is a standard procedure that combines posterior instrumentation with bone grafting to enhance fusion [5]. Pseudoarthrosis or non-union diagnosed ≥1-year post-operatively is the main cause of fusion failure in spine surgery [4,6]. The rate of pseudarthrosis has been reported to be 0–3% with either allograft or autograft bone [4,7].

Autologous iliac crest bone grafts have long been the gold standard in posterior spine fusion [3,8,9]. However, iliac bone harvesting is associated with increased surgical time and may lead to donor site morbidity, with a risk of infection and loss of sensation or chronic pain [6,10,11]. In addition, the quantity and properties of available autologous grafts are limited. Different types of bone substitutes have been used as alternatives to autologous grafts, including allografts, ceramics, and synthetic bone substitutes [9,12]. Allografts are not free of viral contamination, and their availability is limited [10,13]. Synthetic bone substitutes have variable results but are convenient for the surgeon, easily resourced and ready to use [6].

Different bone substitutes are available on the market, but the data are limited, and no compound has yet proven to be superior to others [14]. However, 45S5 bioactive glass is an innovative biomaterial composed of optimal proportions of silicon, calcium, sodium, and phosphorus minerals. Published reports have confirmed its safety and efficacy in various adult orthopaedic conditions and procedures. The use of novel biomaterials in paediatric patients is always a concern in terms of tolerance and efficacy, particularly in posterior spinal fusion, where a large amount of graft material is needed. A study conducted by Ilharreborde et al. in 2008 [9] suggested that bioactive glass can be used in place of autologous grafts as an effective bone substitute in AIS. The safety and efficacy of bioactive glass in paediatric spinal deformities have not yet been evaluated, but there was no significant loss of correction between the 1st erect radiograph and the 24-month post-operative radiograph. There was no sign of non-union, implant displacement or rod breakage.

In our clinical practice, we routinely use bioactive glass to enhance fusion in scoliosis patients. The aim of this retrospective study was to evaluate and compare the post-operative safety and efficiency of bioactive glass 45S5 putty and granules in posterior spine fusion for AIS and NS in a paediatric cohort.

## 2. Materials and Methods

This study was conducted in accordance with the Declaration of Helsinki and the current regulations and reference methodology between July 2018 and December 2022 in a single institution. The study was approved by the Institutional Review Board CPP Ile de France 2 on 07/20/2020: No. ID RCB: 2020-A01071-38. An information letter was sent to all patients and their guardians. The present study was conducted according to the Strengthening the Reporting of Observational Studies in Epidemiology (STROBE) statement [15].

### 2.1. Patient Selection

The inclusion criteria were as follows:-paediatric patient < 20 years old (AIS or NS);-scoliosis requiring posterior fusion posterior instrumentation;-use of bioactive glass (Glassbone Granules or Glassbone Injectable Putty, NORAKER, Lyon-France) as adjuvant fusion;-minimum of 2 years of follow-up.-The exclusion criteria were as follows:-surgical revision;-patient opposition to data collection.

### 2.2. Surgical Technique

All procedures were performed by the same surgeon. A classic straight dorsal incision, centred on the patient’s spinous processes, was performed. The posterior vertebral arch was then exposed. Hybrid constructs, which combine screws, sublaminar bands and hooks, were typically used in addition to cobalt-chrome 6 mm rods. A combination of different correction manoeuvres was performed, including rod rotation, postero-medial translation and in situ contouring. A typical construct is depicted in Figure 1.

In all patients, bioactive glass in the form of GlassBone Injectable Putty or Granules (NORAKER—Lyon/France) was applied to the spine after facetectomies and standard decortication of the laminae at the end of the procedure. GlassBone Granules are composed of 45S5 bioactive glass. GlassBone putty is an injectable paste composed of 45S5 bioactive glass granules mixed with an absorbable binder combining polyethylene glycol and glycerol. The choice between putty or granules relied only on the availability of the putty on the market. Granules came first on the market and putty second. The bone harvested from the facetectomies, and the spinous processes was not used for additional grafts.

### 2.3. Outcomes of Interest

Baseline demographic data such as gender, age at surgery, skeletal maturity (Risser grade) and Lenke curve type were collected.

The occurrence of any anomaly and/or complication was recorded at each post-operative visit (15 days, 6 months and 2 years). Postoperative radiographs were performed at each follow-up and were evaluated for instrumentation failure, bone fusion and Cobb angle. Bone fusion in the instrumented section was classified as acquired, in progress or not acquired. Cobb angle measurement was performed at post-operative discharge and at the final follow-up visit, and the results were compared [9]. Pseudarthrosis was defined as a loss of correction manifested as a difference of >10° between the immediate and final post-operative measurements [7,16]. As it is now accepted that loss of correction after fusion in AIS usually occurs within 2 years after the procedure [17], we used the same time interval for our study. Any screw loosening was also reported.

Pre- and post-operative radiological evaluations were performed using the EOS system (EOS-Imaging—Paris, France). EOS is a low-dose imaging system providing simultaneous AP and lateral views in a stand-up position [18,19]. Semiautomatic 3D reconstruction, using SterEOS software (EOS-Imaging—Paris, France), is based on identifiable anatomic points [20,21]. It provides a 3D image of the spine deformity, giving measurements of spine parameters in a stand-up position. The spine 3D geometry is limited between T1 and S1 since cervical spine is not routinely captured. Validation of the accuracy and reproducibility of the 3D reconstruction method has been reported in previous studies [20,22,23]: the 95% prediction limits for the intra- and inter-observer errors in measurement were computed. The 95% prediction limits indicate the difference between two successive replicate measurements that would exceed approximately 5% of the time due to an error of measurement. The inter-observer 95% prediction for the Cobb angle was 2.8°. The intra-observer 95% predication for the Cobb angle was 2°.

### 2.4. Statistical Analysis

All included patients were considered in the evaluation. A descriptive analysis of all variables of interest was performed. Ellistat (version 5.31; 2020/04, France) was used to perform t tests and other statistical tests. Continuous data are expressed as the mean and standard deviation, while categorical variables are expressed as percentages. Student’s *t*-tests or Mann–Whitney U tests were used to compare the mean pre- and post-operative measurements. The qualitative variables are presented as counts and frequencies. The 95% confidence intervals and statistical significance are presented when relevant. The primary endpoint was the rate of adverse events at least 1 year after surgery.

## 3. Results

### 3.1. Patient Selection and Demographic Data

A total of 43 children and adolescents were included retrospectively (30 females, 69.8%, and 13 males, 30.2%); their mean age at the time of surgery was 15.4 years (range 11–19 years). A flowchart of the study is presented in Figure 2.

Patient demographic and clinical data are recorded in Table 1. Each patient’s last follow-up was performed at 24 months after the surgery.

### 3.2. Peri-Operative Data

All patients underwent posterior thoracolumbar spinal fusion. The average number of instrumented vertebrae was 10 ± 3 [4–15], with 62.8% of patients having more than ten levels instrumented. Detailed peri-operative data are presented in Table 2. The mean operative time was 202 ± 66 [90–300] min. In the putty group, all patients received 20 cc of Glassbone injectable putty (NORAKER—Lyon/France); in the granule group, 14 (78%) patients received 10 cc and 4 (22%) received 20 cc of Glassbone granules (NORAKER—Lyon/France) without adjuvant. The mean hospital stay was 6 ± 3 days [4–15].

### 3.3. Safety

Four of the 43 operated patients experienced adverse events. Three complications appeared early during immediate post-operative follow-up. Two patients had surgical site infection (4.7%), which was treated with revision and cleaning, and one patient had an extended stay in the intensive care unit (2.3%). All these adverse events were due to surgical intervention. No other causes were identified. One case (2.3%) of late mechanical complications was observed 24 months after surgery. The patient was diagnosed with proximal junctional kyphosis (PJK) with dislocation of the proximal hooks; surgical revision was performed, and the instrumentation was removed. No other complications were observed during follow-up.

### 3.4. Radiographic Analysis

The results from the radiographic measurements are summarized in Table 3. At the latest follow-up, bony fusion was documented in all patients. The radiographic parameters of the two groups at each follow-up are presented in Table 3.

The mean pre-operative Cobb angle was 62.7° [30°–130°], and the mean Cobb angle at the 24-month follow-up was 27.1° [0°–70°]. There was a significant difference between the pre-operative and post-operative measurements (Figure 3. This change reflected a significant reduction in spinal deformity.

The mean post-operative Cobb angle on the 1st X-ray (after hospitalization) was 26.5° [0°–68°]. No significant loss of correction occurred between the immediate post-operative examination and the 24-month follow-up. There was no sign of non-union, screw loosening, implant displacement or rod breakage.

## 4. Discussion

The main finding of the present study was that bioactive glass, both in putty and in granular form, is efficient and safe to use in association with proper instrumentation, facetectomies and posterior arch decortication to enhance posterior fusion in young patients with adolescent or neuromuscular scoliosis as evaluated 2 years after surgery.

The clinical and radiological characteristics of our cohort, along with the surgical procedure and the rate of revisions and complications, are in line with the results obtained in other recent studies [9,24,25]. None of the observed patients experienced a post-operative increase in the Cobb angle by >10°, indicating that bioglass alone is sufficient to promote fusion.

Iliac crest graft represented the gold standard for many years, but they are known to be associated with donor site morbidity [3,6,8,9,11,12]. Furthermore, the grafts may be harvested in insufficient quantity for patients requiring long fusion. At present, different synthetic options are available to surgeons, and many have proven to be as effective as iliac crest grafts [4]. These biologic materials allow solid fusion while reducing the surgical time and eliminating the donor site morbidity associated with iliac crest grafting. Ilharreborde et al. [9] reported that the use of bioactive glass in addition to local autologous bone grafts in AIS was as effective as autologous iliac crest bone alone. To the best of our knowledge, the present study is the first to show that the use of bioglass alone also represents a viable and safe option for enhancing fusion in scoliosis surgery.

While CT scans would represent the most reliable tool to evaluate the fusion mass, this imaging technique is not routinely used at our institution to limit radiation exposure [26]. Therefore, we performed an indirect evaluation of the fusion rate using the definition of pseudarthrosis suggested by Price et al. [7]. A loss of correction measuring more than 10° of Cobb angle over the observation period was taken to define a non-fused spine [7]. The mean loss of correction was less than 2° in our series, which is within the accepted 3° measurement error (Figure 4).

The need for a safety evaluation of the massive use of bioglass on the spine is evident from the issues experienced with high doses of bone morphogenetic proteins (BMPs) in spine surgery [27]. Bioactive glass is an osteoconductive bone substitute and not an osteoinductive agent as BMP is, meaning that bioglass merely acts as a scaffold to promote the settlement of osteoblasts arising from bone decortication. BMPs create bone in a bone-free environment and have a well-documented dose effect. As safety is a priority and a legitimate concern when applying newly developed biomaterials in the human body, we kept this concern in mind and examined safety as an outcome of this study. In the cohort that we observed, bioactive glass did not have the disadvantage of a dose effect. At least 20 cc of bioactive glass was applied in most of the patients without adverse effects. While a longer follow-up will be required to investigate possible long-term effects, we believe that, to the osteoconductive rather than osteoinductive nature of bioglass, there will not be long-term complications associated with the use of this material.

In studies on oral microorganisms in vitro, bioglass has demonstrated antibacterial properties, which may reduce the potential for bacterial colonization of the grafted sites [28,29]. The 4.7% infection rate in our study is equivalent to the values reported in the recent literature. Both patients who developed wound infections in this study were NS patients, and people with this condition are known to be more prone to infections than people with AIS. We were unable to evaluate the antibacterial properties owing to the design of the study and the small sample of patients, and we did not detect a trend in the rate of operating site infections in our patients to support these properties. However, in light of our data compared with the literature, it is highly unlikely that the observed wound infections were connected to the use of bioglass.

We did not observe a significant difference in outcomes between patients who received putty grafts and those who received granular grafts. The bioactive glass putty plays the same role as granules. There are no primary mechanical properties to consider when applying bioactive glass putty. This is not an issue in posterior spinal fusion because the instrumentation assures primary mechanical stabilization, but a putty graft may not be a suitable stand-alone solution for bone filling. However, the “wet sand” consistency and adhesive properties facilitate the accurate placement of the biomaterial. Once applied to the bone, it does not move, even in the event of irrigation or bleeding. It is also very useful in intersomatic cages. Hammering a cage during insertion does not dislodge the putty from the cage, as is usually experienced when small autologous bone fragments are used instead. For those reasons, bioactive glass putty has progressively replaced granules in most spine procedures.

The retrospective nature and uncontrolled design of this study are its main limitations. While these observations confirmed the efficacy and safety of stand-alone bioactive glass 45S5 as an alternative to autologous bone grafts, further studies will be required to compare the available materials and assess possible differences among the various compounds.

## 5. Conclusions

PSF is currently a common procedure that has a very low rate of complications, regardless of the type of biomaterial used. Bioactive glass in the form of putty or granules is an easily handled biomaterial but still a newcomer on the market. This study shows that its massive use in posterior fusion, when combined with proper surgical planning, hardware placement and correction, is effective in providing good clinical and radiological outcomes.

## Figures and Tables

**Figure 1 children-10-00398-f001:**
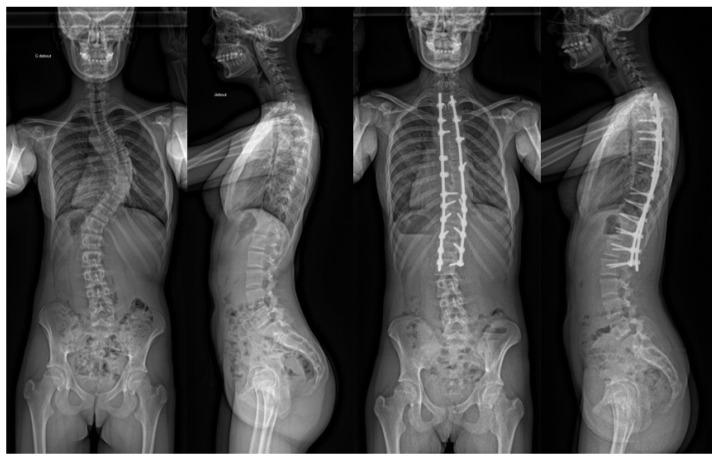
Pre- and post-operative full-spine coronal and sagittal X-rays illustrating a typical long construct for posterior fusion to correct a deformity.

**Figure 2 children-10-00398-f002:**
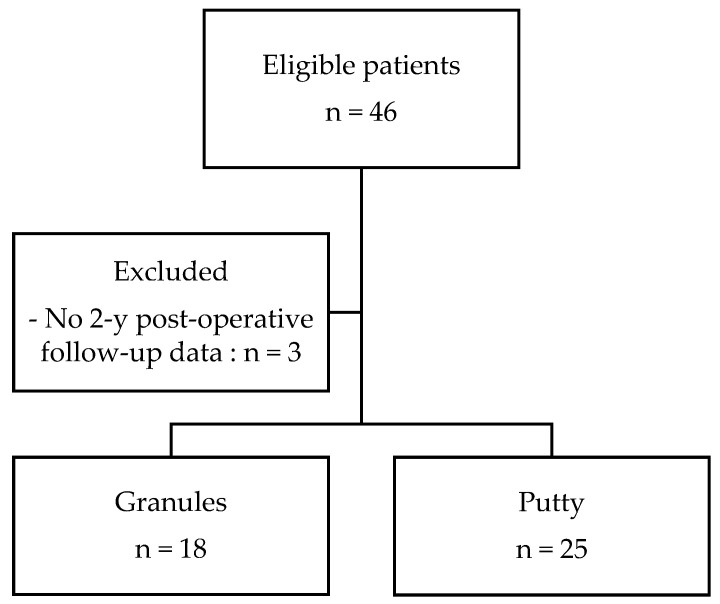
Flowchart of the study.

**Figure 3 children-10-00398-f003:**
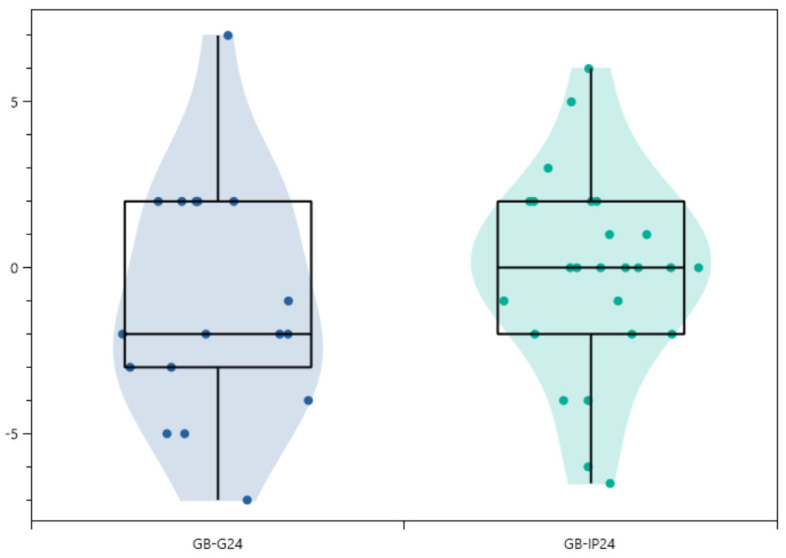
Box plots of loss of correction in the granule group (GB-G24) and the putty group (GB-IP24) groups, 24 months follow-up. There were no outliers. More than 50% of the data are included in the box plot for each material. The median is 0 for the putty group and −2 for the granule group. The results reflect little or no loss of correction.

**Figure 4 children-10-00398-f004:**
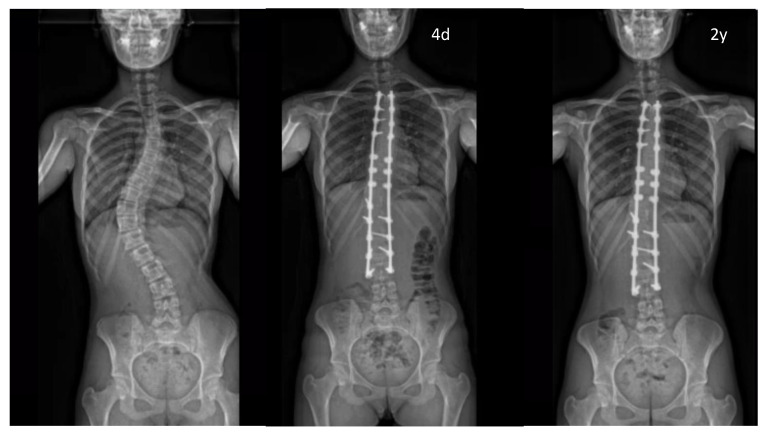
Pre- and post-operative full-spine coronal X-rays (1st erect and 2y po) illustrating the proper stabilisation of the main curve (<10°) without any screw loosening. (4d: Day 4 after surgery).

**Table 1 children-10-00398-t001:** Patient characteristics. Comparisons were computed between the granule and putty groups. There was no significant difference between the 2 groups.

Characteristic		N = 43		Granules (*n* = 18)		Putty (*n* = 25)		*p* Value between Groups
Age (years), mean ± SD		15.4 ± 1.9 [11–19]		15.7 ± 1.7 [13–19]		15.2 ± 2.0 [11–19]		*p* = 0.466—NS
	Female	30 (69.8%)		10 (55.6%)		20 (80%)		/
	Male	13 (30.2%)		8 (44.4%)		5 (20%)		/
Weight (kg)		49.4 ± 9.9 [31–77]		47.9 ± 11.5 [31–71]		50.4 ± 8.7 [37–77]		*p* = 0.413—NS
Size		1.60 ± 0.06 [1.50–1.75]		1.58 ± 0.07 [1.50–1.70]		1.61 ± 0.06 [1.50–1.75]		*p* = 0.402—NS
BMI (kg/m^2^)		19.9 ± 3.3 [15.8–29.7]		20.8 ± 3.6 [15.8–26.1]		19.5 ± 3.2 [16.0–29.7]		*p* = 0.290—NS
Smoking		None		/		/		/
Indication								
Adolescent idiopathic scoliosis		34 (79.1%)		9 (50%)		25 (100%)		/
Neurologic scoliosis		7 (16.3%)		7 (38.9%)		/		/
Neuromuscular scoliosis		2 (4.7%)		2 (11.1%)		/		/
Lenke classification		1A 2A 1B 3C 5C 1C	20 (46.5%) 2 (4.7%) 3 (7.0%) 1 (2.3%) 16 (37.2%) 1 (2.3%)	1A 2A 1B 3C 5C 1C	6 (33.3%) 0 (0%) 2 (11.1%) 0 (0%) 9 (50%) 1 (5.6%)	1A 2A 1B 3C 5C 1C	14 (56%) 2 (8.0%) 1 (4.0%) 1 (4.0%) 7 (28%) 0 (0%)	/
Risser classification		1 2 3 4 5	2 (4.7%) 3 (7.0%) 3 (7.0%) 30 (69.8%) 5 (11.6%)	1 2 3 4 5	2 (11.1%) 2 (11.1%) 2 (11.1%) 11 (61.1%) 1 (5.6%)	1 2 3 4 5	0 (0.0%) 1 (4%) 1 (4%) 19 (76%) 4 (16%)	/

**Table 2 children-10-00398-t002:** Distribution of the number of instrumented levels.

	N (%)	Granules (%)	Putty (%)
Mean number of levels	10 ± 3 [4–15]	12 ± 3 [5–15]	8 ± 3 [4–12]
Number of instrumented levels			
≥10	27 (62.8%)	16 (88.9%)	11 (44%)
8–9	7 (16.3%)	1 (5.6%)	6 (24%)
6–7	0 (0%)	0 (0%)	0 (0%)
≤5	9 (20.9%)	1 (5.6%)	8 (32%)

**Table 3 children-10-00398-t003:** Radiographic data (Cobb angle, correction rate, loss of correction). NS: Not Significant.

	N = 43		*p* Value	Granules (n = 18)		*p* Value	Putty (n = 25)		*p* Value
	Mean (n)	Range		Mean	Range		Mean	Range	
Cobb angle									
Pre-op	62.7 ± 22.7 (43)	[30–130]	/	70.4 ± 24.9 (18)	[42–130]	/	57.2 ± 19.6 (25)	[30–120]	/
1st erect	26.5 ± 16.4 (43)	[0–68]	*p* < 0.05 from pre-op (7.10−13)	30.1 ± 17.9 (18)	[2–68]	*p* < 0.05 from pre-op (3.10−6)	23.9 ± 15.1 (25)	[0–50]	*p* < 0.05 from pre-op (2.10−8)
3–6 months	24.0 ± 13.9 (25)	[0–50]	*p* < 0.05 from pre-op (8.10−11)	23.0 ± 7.1 (2)	[18–28]	*p* < 0.05 from pre-op (0.02)	24.1 ± 14.5 (23)	[0–50]	*p* < 0.05 from pre-op (3.10−8)
24 months	27.1 ± 16.1 (42)	[0–70]	*p* < 0.05 from pre-op (10−12)	31.2 ± 18.6 (17)	[7–70]	*p* < 0.05 from pre-op (9.10−6)	24.1 ± 14.1 (25)	[0–52]	*p* < 0.05 from pre-op (10−8)
Correction rate									
Pre-op vs. 1st erect (°)	36.2 ± 12.0 (43)	[15–70]	/	40.3 ± 11.4 (18)	[24–70]	/	33.2 ± 11.7 (25)	[15–70]	/
Loss of correction									
1st erect vs. 3/6 months (°)	−0.55 ± 3.32 (25)	[−8.0–5.0]	*p* = 0.874—NS	0.00 ± 5.7 (2)	[−4.0–4.0]	*p* = 0.950—NS	−0.60 ± 3.25 (23)	[−8.0–5.0]	*p* = 0.773—NS
1st erect vs. 24 months (°)	−0.65 ± 3.24 (42)	[−7.0–7.0]	*p* = 0.671—NS	−1.12 ± 3.52 (17)	[−7.0–7.0]	*p* = 0.685—NS	−0.18 ± 2.97 (25)	[−6.5–6.0]	*p* = 0.868—NS

## Data Availability

Data available on request to corresponding author.

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
