# Peer review of "Safety and Efficacy of Stand-Alone Bioactive Glass Injectable Putty or Granules in Posterior Vertebral Fusion for Adolescent Idiopathic and Non-Idiopathic Scoliosis"

_children, 2023, doi:10.3390/children10020398_

Round 1
Reviewer 1 Report
I encourage Authors to provide the exact formula of the bioactive glass, which has been used since there are different bioactive glass providers in the market and results can not be representative of all kinds of bioactive glass products in the market.
Otherwise clear and concise well written article.
Author Response
GlassBone Granules are composed of 45S5 bioactive glass. GlassBone putty is an injectable paste composed of 45S5 bioactive glass granules mixed with an absorbable binder combining polyethylene glycol and glycerol.Reviewer 2 Report
This article focused on the usefulness of bioactive glass in posterior fusion for correction surgery of AIS. There are several problems about this article.
The main goal of this article is to show the effectiveness of bioactive glass, however, the authors divided the subjects into two groups. Furthermore, the selection criteria (Putty or granules) are not clearly mentioned. How did authors choose the use of bioactive glass with putty or granules?
The fusion length tends to longer in non-idiopathic scoliosis. The bone quality may be different between idiopathic scoliosis and neuromuscular scoliosis. Why did authors include many pathologies of scoliosis?
The criterion of fusion(pseudoarthrosis) is not suitable for this study. To be sure, the change of Cobb angled of 10 degrees may not change patients outcomes. But, this article focused on the usefulness of bioactive glass without autologous or artificial bone graft or BMPs. If this article shows the bioactivity of bioactive glass, the evaluation of fusion or pseudoarthrosis should be stricter method. Who measured the Cobb angles? How is the interobserver and intraobserver reliability? Why not examined the loosening of all pedicle screws? Why did the authors measure only the whole Cobb angles and not the Cobb angles of each segment?
The bony union under the use of bioactive glass should be shown in CT scans or other image modalities in Discussion section.
Author Response
Responses to reviewer 2
This article focused on the usefulness of bioactive glass in posterior fusion for correction surgery of AIS. There are several problems about this article.
The main goal of this article is to show the effectiveness of bioactive glass, however, the authors divided the subjects into two groups. Furthermore, the selection criteria (Putty or granules) are not clearly mentioned. How did authors choose the use of bioactive glass with putty or granules?
R: As mentioned in the discussion, granules were first available on the market. Therefore, we used granules in our earlier practice with bioglass. Glassbone putty came afterward on the market, its usability is better than granules for posterior spine fusion. We use putty in most of our cases now. We’ve added this point in M&M.
The fusion length tends to longer in non-idiopathic scoliosis. The bone quality may be different between idiopathic scoliosis and neuromuscular scoliosis. Why did authors include many pathologies of scoliosis?
R: The main idea of this study was to evaluate the safety first and efficacy of bioactive glass. As you mention bone quality may be very different in neuromuscular scoliosis. This placed the bioglass evaluation in the worst case. We thought it could be interesting to evaluate the bone substitute in cases where we use it the most. Moreover, the combination of the 2 groups increased the series.
The criterion of fusion(pseudoarthrosis) is not suitable for this study. To be sure, the change of Cobb angled of 10 degrees may not change patients’ outcomes. But this article focused on the usefulness of bioactive glass without autologous or artificial bone graft or BMPs. If this article shows the bioactivity of bioactive glass, the evaluation of fusion or pseudoarthrosis should be stricter method.
R: We agree that 10° of difference may not change the patients’ outcomes. It is however a valid and published method to indirectly evaluate fusion.
Who measured the Cobb angles?
R: The Cobb angle was measured pre and post-operatively on the 3D reconstructions obtained with the EOS system which provides 3° interobserver reliability. This method was added in M&M.
How is the interobserver and intraobserver reliability?
R: 3° on the 3D Cobb angle computed out of the EOS 3D reconstructions.
Why not examined the loosening of all pedicle screws?
R: The loosening of pedicles screw was examined. No loosening was observed at 2 y f/u.
Why did the authors measure only the whole Cobb angles and not the Cobb angles of each segment?
R: The purpose of the study was not to evaluate the global correction. According to our main criteria to evaluate fusion, we needed to measure the Cobb angle on the main curve.
The bony union under the use of bioactive glass should be shown in CT scans or other image modalities in Discussion section.
R: Since we do not perform CT scans postoperatively to prevent patients’ radiation exposure, we added and another patient with X rays at 2 y f/u in the discussion.
Reviewer 3 Report
Scoliosis (PSF) is a standard procedure for treating severe scoliosis, combining bone grafts and/or bone replacements with posterior instrumentation to enhance fusion. These authors evaluated and compared the postoperative safety and efficiency of standalone bioactive glass putty and granules in posterior spinal fusion for scoliosis in pediatric cohort, and conducted a follow-up study at 24 months to verify the results. They confirmed that there was no significant correction loss between the immediate postoperative time and 24-month follow-up. There were also no signs of binding, implant displacement, or rod breakage. The study argues that large-scale use of bioactive glass in posterior fusion is effective in providing good clinical and radiation results when appropriate surgical planning and well-positioned hardware combined with calibration. I think their research will have a great impact on performing PSF through a new method. Although there is not much data presented, I think it can be interpreted in the thesis without much difficulty. In addition, as a result of discussing not only with me but also with orthopedic surgeons working directly in this field, it is judged to be sufficiently valuable as a thesis. Therefore, this reviewer judges that the formet is sufficient as it is now without much difficulty. If there is any request from other reviewers, I would like to revise that part.
Author Response
Thanks a lot for your comments.
Round 2
Reviewer 2 Report
This paper is now acceptable fashion.